# In vivo evolution of an emerging zoonotic bacterial pathogen in an immunocompromised human host

A. Launay[1], C.-J. Wu[1], A. Dulanto Chiang [1], J.-H. Youn[2], P. P. Khil[1,2] & J. P. Dekker [1,2 ✉]

Zoonotic transfer of animal pathogens to human hosts can generate novel agents, but the genetic events following such host jumps are not well studied. Here we characterize the mechanisms driving adaptive evolution of the emerging zoonotic pathogen *Bordetella hinzii* in a patient with interleukin-12 receptor β1 deficiency. Genomic sequencing of 24 *B. hinzii* isolates cultured from blood and stool over 45 months revealed a clonal lineage that had undergone extensive within-host genetic and phenotypic diversification. Twenty of 24 isolates shared an E9G substitution in the DNA polymerase III ε-subunit active site, resulting in a proofreading deficiency. Within this proofreading-deficient clade, multiple lineages with mutations in DNA repair genes and altered mutational spectra emerged and dominated clinical cultures for more than 12 months. Multiple enzymes of the tricarboxylic acid cycle and gluconeogenesis pathways were repeatedly mutated, suggesting rapid metabolic adaptation to the human environment. Furthermore, an excess of G:C > T:A transversions suggested that oxidative stress shaped genetic diversification during adaptation. We propose that inactivation of DNA proofreading activity in combination with prolonged, but sub-lethal, oxidative attack resulting from the underlying host immunodeficiency facilitated rapid genomic adaptation. These findings suggest a fundamental role for host immune phenotype in shaping pathogen evolution following zoonotic infection.

[1] Bacterial Pathogenesis and Antimicrobial Resistance Unit, LCIM, NIAID, Bethesda, MD, USA. [2] Department. Laboratory Medicine, NIH Clinical Center, Bethesda, MD, USA. ✉email: john.dekker@nih.gov

The study of chronic infections in patients with genetic immunodeficiencies has provided powerful insights into the processes underlying bacterial pathoadaptation[1–4]. Most prior work has focused on the within-host evolution of established human pathogens, where the dominant evolutionary selection pressures include antibiotic stress and the host immune response. These studies have illustrated that many general evolutionary processes can be discerned under conditions of within-host adaptation, including genetic diversification of lineages with purifying selection, clonal succession events, and balanced fitness trade-offs[1–5]. In some cases, elevated mutation rates due to evolved mismatch repair (MMR) deficiencies or mobile genetic element insertions have been shown to contribute to rapid evolution[4,6]. Dramatic examples of genome compaction due to pseudogene formation and large chromosomal deletions have also been described, in some circumstances resulting in loss of pathogenicity and evolution of an apparently commensal pathogen–host relationship[3,6]. An important insight from these studies of within-host adaptation has been that genetic modifications that occur during chronic infection often parallel those that underlie the emergence of human-restricted pathogens from broad-host range generalists over longer evolutionary periods[7–10].

*Bordetella hinzii* is a member of β-proteobacteria originally characterized in the respiratory tracts of poultry[8,11–13]. Though primarily associated with nonhuman hosts, *B. hinzii* is a genetic relative of *Bordetella pertussis*, the cause of pertussis in humans[14]. More recently, *B. hinzii* has been recognized to be capable of causing disease in humans following the zoonotic transfer, including respiratory tract infection, cholangitis, bacteremia, and endocarditis[13,15–21]. Based on these reports, *B. hinzii* is considered to represent an emerging pathogen with zoonotic risk to humans, and warrants careful study given the ability of its genetic relatives to cause human epidemics with sustained human-to-human transmission.

In this work, we have studied the adaptation of *B. hinzii* in a patient with recessive interleukin-12 receptor beta 1 (IL-12Rβ1) deficiency. In persons with autosomal recessive IL-12Rβ1, the immune response to intracellular pathogens is impaired in part due to reduced levels of IL-12/IL-23 stimulated interferon-γ

(IFN-γ) production[22,23]. Consequently, one of the hallmark features of this disease is an elevated susceptibility to infections with certain intracellular pathogens, including nontuberculous mycobacteria and *Salmonella*[23]. A previous study that examined the evolution of a chronic intravascular *Salmonella enterica* infection within an IL-12Rβ1 deficient individual demonstrated the emergence of an elevated spontaneous mutation rate due to MutS inactivation and functional genome compaction through the generation of pseudogenes[3]. The work presented here examines a rare case of directly observed adaptation of an emerging zoonotic pathogen to a human host and provides key insights into the immunobiology of host–pathogen interactions in the context of compromised host immunity.

## Results

### *B. hinzii* isolates cultured from an IL-12Rβ1 deficient patient over 45 months demonstrate remarkable phenotypic diversity.

A patient with recessive IL-12Rβ1 deficiency (homozygous for IL12RB1 c.94 C > T p.Gln32Ter) and previously diagnosed chronic *B. hinzii* infection presented to the NIH Clinical Center for treatment. The patient had a history of *Candida tropicalis* esophagitis but no documented history of either mycobacterial infection or salmonellosis. The patient additionally had a history of clinical colitis and membranoproliferative glomerulonephropathy with nephrotic syndrome. During the 45-month course of care that followed, serial blood and stool cultures recovered *B. hinzii* isolates displaying a remarkable degree of phenotypic diversity (Fig. 1, Supplementary Fig. 1, and Supplementary Table 1, 2). The cultured isolates generated morphologies ranging from discrete colonies to confluent mucoid pools, with wide variation in growth rate and antibiotic susceptibility profiles. Furthermore, some of these different morphotypes were admixed together within single colonies in the primary culture, suggesting either that the underlying colony forming units (CFUs) consisted of multicellular aggregates of distinct genetic sub-lineages, or that the CFUs were single cells capable of rapidly diversifying into the observed morphologies. These mixed colonies were separated into morphologically pure subcultures following serial isolation

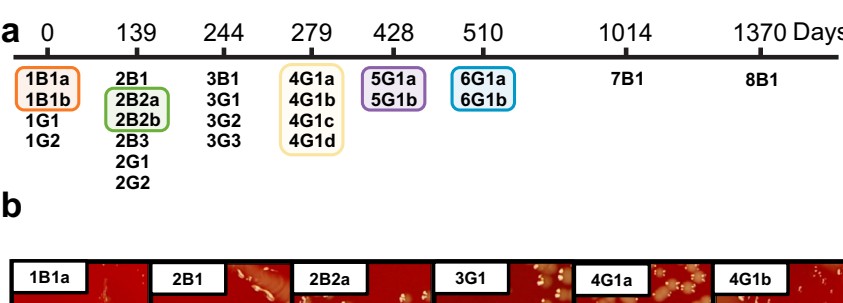

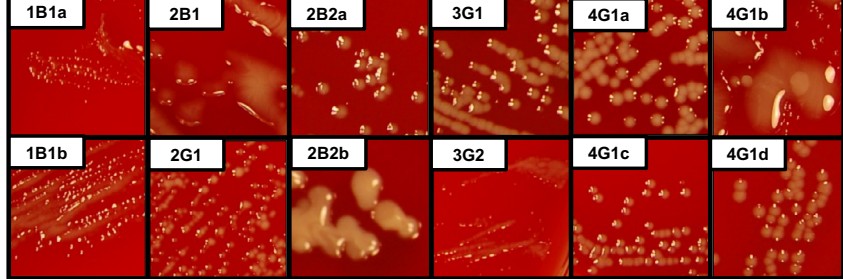

**Fig. 1 Cultured *B. hinzii* isolates demonstrate remarkable phenotypic diversity. a** Timeline of isolate collection. Isolates were named using the following rules: Sequential clinical culture number (1–8); followed by culture source (B blood, G gastrointestinal); followed by single colony number from the source culture (1–3); followed by a letter designating individual morphotypes separated from the single mixed-morphotype colony (a–d). Isolates separated from mixed-morphotype colonies are highlighted by different colors, which is maintained in Fig. 2 for ease of comparison. **b** Examples of the diversity of morphologies observed, which included medium-to-large-sized discrete colonies (3G1, 4G1a) and mucoid pools that ranged from translucent (2B1) to opaque (2B2b).

**Table 1 Ancestral substitutions common to all highly divergent isolates.**

| Position | Ancestor | Variant | Effect | Gene | Locus tag | Annotations |
|---|---|---|---|---|---|---|
| 50562 | T | G | Q502P | tdhA | A_00046 | TonB-dependent heme receptor A |
| 155257 | A | C | Q573P | nhaP2_1 | A_00148 | Putative K(+)/H(+) antiporter NhaP2 |
| 450540 | T | A | H182L | cheD | A_00440 | Chemoreceptor glutamine deamidase CheD |
| 581429 | A | T | I287F | NA | A_00560 | Hypothetical protein |
| 642615 | A | G | E9G | dnaQ | A_00619 | DNA polymerase III subunit epsilon |
| 1386539 | C | A | D238Y | envZ | A_01279 | Homolog of RisS, two-component histidine kinase protein |
| 1402221 | T | G | L65R | gstB_2 | A_01291 | Putative Glutathione S-transferase GST-6.0 |
| 1422293 | C | T | V182M | NA | A_01313 | Hypothetical protein |
| 1539903 | G | A | L212L | sucA | A_01422 | 2-oxoglutarate dehydrogenase E1 component |
| 2064888 | T | A | Q304L | fliF | A_01913 | Flagellar M-ring protein |
| 2101905 | A | G | I262T | ppsR | A_01952 | Phosphoenolpyruvate synthase regulatory protein |
| 2150968 | T | G | Q80P | NA | A_02012 | Hypothetical protein |
| 2222195 | AC | A | frameshift | NA | A_02072 | Hypothetical protein |
| 2459163 | A | T | T37S | icd_1 | A_02295 | Putative Isocitrate dehydrogenase [NADP] |
| 2520180 | T | C | Q112R | yhhQ | A_02350 | Queuosine precursor transporter |
| 3093668 | C | A | P125P | gsiD_5 | A_02912 | Putative Glutathione transport system permease protein GsiD |
| 3296026 | A | T | L53Q | ygaZ | A_03095 | Inner membrane protein YgaZ |
| 3297833 | A | G | I73V | NA | A_03098 | Hypothetical protein |
| 3486692 | A | C | D437A | NA | A_03291 | Putative formate dehydrogenase |
| 3646109 | C | T | - | - | - | intergenic region |
| 3652926 | G | T | A379E | NA | A_03455 | hypothetical protein |
| 3728264 | A | T | V44E | kefC_3 | A_03528 | Putative Glutathione-regulated potassium-efflux system protein KefC |
| 3897012 | G | A | A18V | NA | A_03713 | Hypothetical protein |
| 4268100 | A | T | W100R | ptsJ | A_04057 | Vitamin B6 salvage pathway transcriptional repressor PtsJ |
| 4293566 | T | C | Y30C | NA | A_04082 | Hypothetical protein |
| 4676024 | T | A | L74Q | lpxL_2 | A_04462 | Putative Lipid A biosynthesis lauroyltransferase |

streaking within three passages, indicating that the underlying initial CFUs were in fact aggregates of distinct genetic populations. In total, 24 distinct isolates were recovered following subculturing.

**Sequencing reveals a clonal lineage with extensive genetic diversification**. To establish the genetic basis of this phenotypic diversity, all 24 subcultured isolates underwent whole genome sequencing (Supplementary Table 3). PacBio SMRT sequencing and assembly generated a single circular chromosome of 4.8 Mb in size with 67% GC content and 99.5% identity to the FDA Argos 621 *B. hinzii* genome (Genbank Accession: CP044059). Phylogenetic comparison with other publicly available *B. hinzii* genomes revealed that the patient isolates formed a distinct clade (Supplementary Fig. 2a), though as many as 2179 substitutions separated the most divergent pairs of genomes (median number of SNVs = 665, minimum = 13, Supplementary Fig. 2b). Comparison of complete genome assemblies demonstrated a large-scale rearrangement in 1G1 that involved a ~500 Kb translocation possibly mediated by rDNA repeats (Supplementary Fig. 3). Nine medium-to-large-sized deletions were also observed, including a 53 Kb deletion in isolate 1G1, and a 38 Kb deletion of phage-like element in isolate 8B1 (Supplementary Table 4).

Although all of the isolates cultured from the patient were clearly related, we noted an extreme range of genetic divergence among them (13-2179 SNVs). We inferred the genomic sequence of a likely last common ancestor (LCA) by substituting the most frequent nucleotide at each position into the complete genome assembly of the 2B3 isolate and inferring the ancestral state at polymorphic positions using a set of 12 unrelated complete *B. hinzii* genomes as an outgroup set (see Methods). To provide independent confirmation of the LCA genome sequence, we used a maximum-likelihood approach to build a second LCA genome and found it to be identical in the common regions corresponding to the core genome of the isolates (Supplementary Fig. 4 and Methods). Using Illumina

short-read sequencing data and this reconstructed ancestor as a reference, we then identified a total of 128 small indels, 6320 SNVs, and 11 complex variants involving two close mutational events (Supplementary Data 1). A deeper examination of the variants revealed the presence of two groups of isolates: a high-divergence group and a low-divergence group. Importantly, all high-divergence isolates contained 26 mutations in common that were absent in the four low-divergence isolates and from the LCA (Table 1). These 26 mutations, therefore, reflect the creation of a high-divergence lineage that coexisted with the original strain for an extended period of time.

Examination of the phylogenetic tree (Fig. 2 and Supplementary Fig 4) revealed an additional unusual observation: lineages derived from individual mixed-morphotype single colonies (e.g., 2B2a and 2B2b) were genetically more similar to each other than to those lineages derived from cultures from other days (e.g., 2B2a and 4G1d). Isolates from the same mixed colony shared an average of 54.4 ± 29.6% of their variants in common compared to 3.4 ± 6.8% shared between isolates not from the same mixed colony. Consequently, the structure of the tree is characterized by a star-like topology separating different culture days, with branching tips relating the isolates from mixed-morphotype colonies from single cultures. Although there are indications of progressive divergence over time, there was not adequate statistical power to allow clear discrimination of temporal signals in the data (Supplementary Fig. 5, 6).

**An E9G mutation in the active site of the ε-subunit of DNA Pol III resulted in a proofreading deficiency driving genomic diversification**. We next sought to understand the basis for the genetic variation observed in the set of high-divergence isolates. One of the 26 mutations separating the high-divergence lineage from the low-divergence lineage is a nonconservative E9G amino acid substitution in the ε-subunit of the major replicative polymerase DNA Pol III. The ε-subunit provides 3′-5′ proofreading

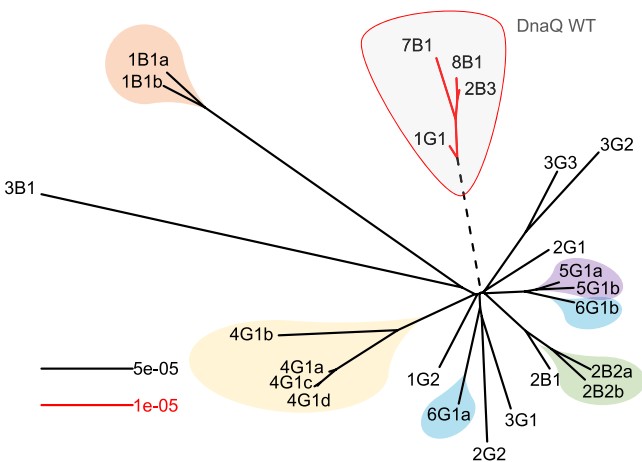

**Fig. 2 Substantial genomic diversity is present in DnaQ E9G isolates.** Unrooted maximum likelihood tree of patient *B. hinzii* isolates constructed using variants relative to the two reconstructed ancestors (see Methods for details and companion rooted tree in Supplementary Fig. 4). Isolates connected by black branches contain an E9G substitution in the ε-subunit of DNA polymerase III (DnaQ E9G), while the isolates connected by red branches carry the wild-type DnaQ allele. The scale of the red branches was magnified five times for clarity of presentation. A prominent feature of the phylogenetic tree is that isolates derived from the same mixed-morphotype single colonies (highlighted by colors corresponding to those used in Fig. 1) are genetically more related to each other that isolates from cultures from different days. Source data are provided as a Source Data file.

exonuclease activity as a part of the holoenzyme and plays a structural role as well, and the glutamate to glycine (E9G) osubstitution falls into the Exo I motif that is critical for this exonuclease activity[24,25]. Mutations within the Exo I motif of the ε-subunit of DNA Pol III result in partial or complete inactivation of proofreading activity in *Escherichia coli*[26–28]. Analysis of available sequence and structural data from other organisms reveals that the E9G substitution eliminates a conserved glutamate residue essential for binding of the catalytic divalent metal ion with its γ-carboxyl group (Fig. 3a, b). Simulations have suggested that in addition to forming a critical part of the catalytic metal-binding site, the glutamate γ-carboxyl group is also involved in proton transfer as a part of the catalytic reaction mechanism and thus critically important for exonuclease activity[29]. We thus hypothesized that proofreading activity was disrupted in the DnaQ E9G isolates, and that the resulting higher mutation rates drove diversification. To evaluate the impact of the DnaQ E9G substitution on spontaneous mutagenesis, we performed in vitro mutation accumulation (MA) experiments. In agreement with predicted DNA Pol III proofreading defects, the spontaneous mutation rate was higher by more than two orders of magnitude in the DnaQ E9G mutants compared to wild-type *B. hinzii* (Fig. 3c). Additionally, the markedly larger genetic divergence of the DnaQ E9G isolates from their likely LCA is concordant with these higher mutation rates (Fig. 3d).

**Mutations common to all proofreading deficient lineages suggests early adaptive selection.** While WT DnaQ isolates were cultured both early and late during the course of infection, the majority of isolates recovered over a period of 16 months belonged to the DnaQ E9G clade, raising the question of why these proofreading deficient hypermutators were so successful. Though elevated mutation rates are usually deleterious given sufficient time for mutational load to accumulate, over the short-term they may be beneficial in the context of infection, where

rapid adaptation to host immune defenses and antibiotics is critical[6,30–32]. Hypermutation due to the DnaQ E9G mutation may therefore have resulted in a selective advantage by facilitating secondary adaptive mutations that were rapidly discovered and selected during the early course of clinical infection. We hypothesized that the DnaQ E9G substitution hitchhiked with one or more of the other initial mutations that conferred a fitness advantage. Indeed, there are 24 coding and one intergenic substitution common to all isolates of the DnaQ E9G clade (Table 1). Twenty-one of these coding mutations were nonsynonymous with mostly nonconservative substitutions, including 7 of 21 substitutions adding or removing a charge, and an additional frameshift mutation. To evaluate the selective pressures on coding sequences, we performed genome-wide dN/dS analysis. When calculated over the whole genome, we obtained a dN/dS ratio of 0.81, suggesting mild purifying selection overall. Strikingly, for the set of 24 coding substitutions in the DnaQ E9G clade founder, the dN/dS ratio was 3.51, significantly higher than the overall genome-wide dN/dS ratio (Fisher's exact test, $p < 0.016$), providing evidence that mutations common to DnaQ E9G lineage were under strong positive selection.

Gene ontology analysis of this set of substitutions indicated an enrichment of genes from the tricarboxylic acid cycle pathway (Fisher's exact test, $p < 0.01$) and a weaker enrichment of genes associated with proton transmembrane transport, phosphorylation-dephosphorylation processes, and cellular metabolic salvage pathways (Fisher's exact test, $p < 0.05$) (Supplementary Fig. 7). Notably, two genes annotated as functionally linked enzymes from the tricarboxylic acid cycle contained nonsynonymous mutations, isocitrate dehydrogenase (*icd_1*) and the phosphoenolpyruvate regulatory protein (*ppsR*). Additionally, a non-synonymous mutation is present in the *B. hinzii* homolog of the two-component histidine kinase sensor protein RisS (annotated as EnvZ in *B. hinzii*). RisS is required for in vivo persistence in *Bordetella bronchiseptica*[33], but appears to be a pseudogene in human-adapted *B. pertussis*[34].

**Mutation rates and spectra cluster into three distinct groups and reveal an impact of oxidative stress.** We observed that many of the DnaQ E9G isolates had accumulated non-synonymous mutations in DNA replication and repair genes, including the DNA polymerase III α-subunit *dnaE;* the error-prone translesion DNA polymerase *dnaE2;* the MMR proteins *mutS* and *mutL;* the base excision repair proteins *mutY and mutM;* and the nucleotide-excision repair proteins *uvrA_1, uvrA_2, uvrB, uvrC,* and *uvrD*, suggesting the possibility of complex compound hypermutator phenotypes. We thus examined mutational processes in these isolates in greater detail. Analysis of the mutations present in the patient isolates revealed a range of spectra that could be clustered into three groups: DnaQ WT, DnaQ E9G, and a third group of three isolates (1B1a, 1B1b, and 3B1) that contained the DnaQ E9G mutation in combination with additional mutations in the base excision repair genes *mutM* and *mutY* (Figs. 4, 5, Supplementary Figs. 8, 9 and Supplementary Table 5). These three isolates demonstrated even higher spontaneous mutation rates, greater divergence from the LCA, and a markedly increased proportion of G:C > T:A transversions, consistent with secondary inactivation of base excision repair[35,36].

In addition, there are finer differences in the mutational spectra. While the spectra of spontaneous mutations seen in vitro largely match the SNV spectra seen in the patient isolates, the mutational spectra of the patient isolates were more tightly clustered (Fig. 5 and Supplementary Fig. 9). This may reflect additional later diversification of mutational mechanisms due to the acquisition of secondary modifier mutations in DNA repair

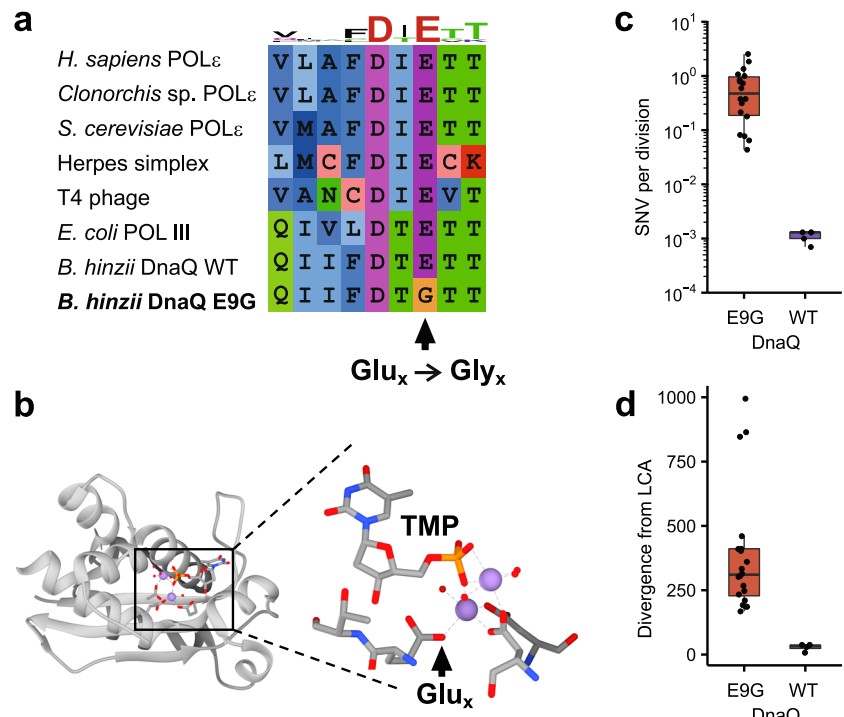

**Fig. 3 E9G substitution in Pol III ε-subunit results in loss of conserved catalytic metal ion binding site and elevated mutation rates. a** Multiple sequence alignment of exonuclease domains from different organisms demonstrates absolute conservation of glutamate mutated in DnaQ E9G protein. **b** Crystal structure of *E. coli* DnaQ (PDB: 2IDO) demonstrates that the γ-carboxyl residue of the DnaQ E9 glutamate (Glu$_x$) is adjacent to thymidine-5′-phosphate (TMP) and forms the binding site for one of the two catalytic divalent metal ions (Mn$^{2+}$, purple) in the active site, critical for the exonucleolytic function. Substitution of glycine in DnaQ E9G eliminates this critical γ-carboxyl group. **c** Spontaneous mutation rate estimated in MA experiments is higher in DnaQ E9G ($n = 18$) compared to WT DnaQ isolates (including both patient and ATCC isolates respectively $n = 2$ and $n = 3$). Horizontal lines inside boxes indicate median values, the upper and lower edges correspond to the 25th and 75th percentiles, and whiskers extend to 1.5× IQR. **d** Genetic divergence from the LCA is higher within the DnaQ E9G isolates ($n = 20$) than within the WT DnaQ isolates ($n = 4$). Horizontal lines inside boxes indicate median values, the upper and lower edges correspond to the 25th and 75th percentiles, and whiskers extend to 1.5× IQR. Source data are provided as a Source Data file.

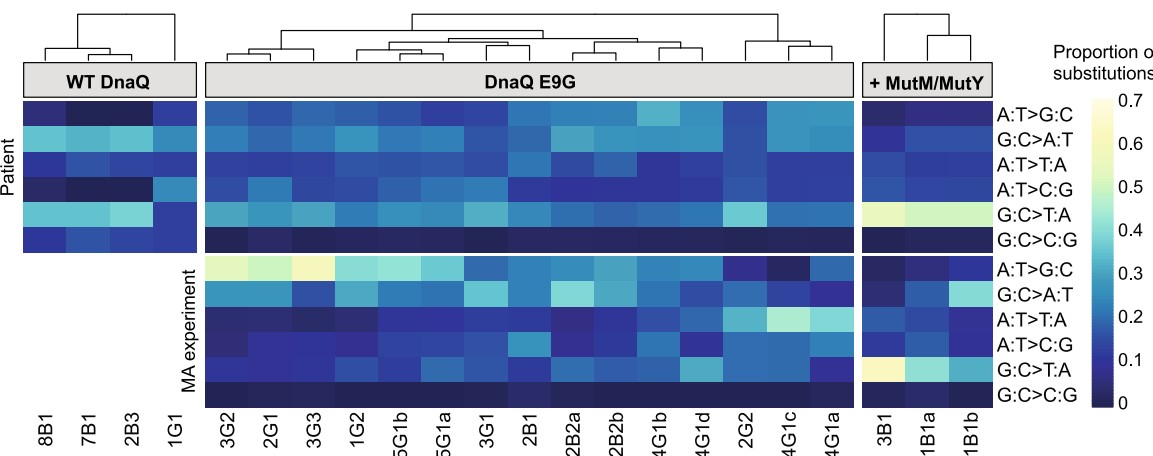

**Fig. 4 Diversity of base-pair substitution spectra in *B. hinzii* isolates is largely explained by mutations in DNA repair pathways.** Heatmap of clustered base-pair substitution spectra in patient isolates (top panel) and measured in mutation accumulation experiments (bottom panel). Relative mutation proportions are indicated by color for each of the base-pair substitutions as shown on the color scale legend (see Supplementary Table 5 for numeric data). Base-pair substitution frequency spectra in WT DnaQ patient isolates, DnaQ E9G isolates, and DnaQ E9G isolates with additional mutations in MutM or MutY genes were clustered using hierarchical clustering and indicated by dendrogram at the top. Mutational spectra detected in patient isolates are similar to those in the mutation accumulation experiments and are strongly affected by the DnaQ and MutM/MutY genotypes. Source data are provided as a Source Data file.

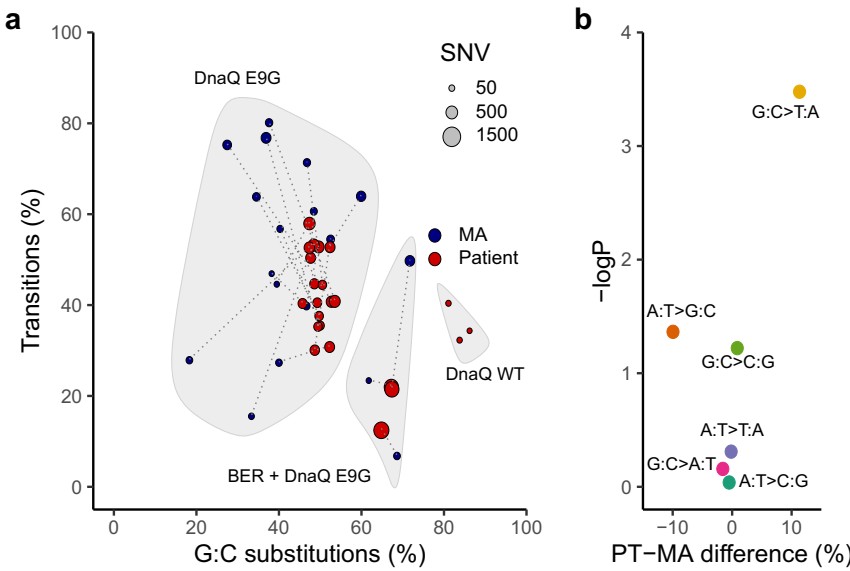

**Fig. 5 Distinct mutation signatures observed in patient isolates reveal impacts of DnaQ and base excision repair pathway genotypes and oxidative stress. a** Relative transition and GC substitution frequencies in patient isolates (red dots) and in mutation accumulation experiments (blue dots). Dot size represents a divergence from an ancestor (SNVs) and matching isolates are connected by a dashed line. Three distinct groups defined by DnaQ and base excision repair (MutM or MutY) mutations are indicated. Isolate 1G1 is not included in the figure due to the small number of mutations. **b** Mean differences in the relative frequency of base-pair substitutions for mutations observed in patient isolates vs. MA experiment. The vertical axis indicates the negative $\log_{10}$ of the calculated $P$ values (two-sided Wilcoxon test) for the indicated base-pair substitution. A significant excess of G:C > T:A substitutions is present in the patient isolates, suggesting that oxidative stress played a major role in shaping in vivo evolution. Source data are provided as a Source Data file.

genes. Strikingly, despite all of the divergence in mutational spectra, there is a significant excess of G:C > T:A transversions (Wilcoxon test, $p = 0.0071$) in the patient isolates compared to MA experiments (Fig. 5 and Supplementary Fig. 10). This may reflect 8-oxoguanine mutagenesis as a consequence of within-host oxidative attack[35].

**Multiple genes are targets of adaptive selection.** To obtain better insight into the selective forces acting on *B. hinzii* during within-host adaptation, we asked whether any genes were more frequently mutated than expected by chance. To estimate the number of mutations expected by chance, we randomly redistributed the nonredundant set of unique SNVs through the genome and found that 134 genes were enriched for mutations (unadjusted $p < 0.05$, Supplementary Data 2, see Methods for details). The most frequently mutated genes were affected in more than 50% of all isolates and contained 10–12 different mutations (Supplementary Data 2). Furthermore, we found that dN/dS ratios were significantly elevated for many of the top mutational targets (Supplementary Data 3 and Methods) suggesting that they were under positive selection. Due to the low number of mutations per gene, however, the statistical power of the dN/dS calculations is limited and should be interpreted with caution. Importantly, both SNV frequency spectra and homoplasy analysis indicate that nearly all of these mutations were independent. A homoplasy test identified 85 mutations in 74 genes (six variants were intergenic) that could not be explained by shared ancestry (Supplementary Data 2). These mutations could be due to true convergent evolution or recombination; alternatively, they may represent artifacts in the tree reconstruction process. Overall, these mutations represented a small percentage of total mutations (1.3%), and all other mutations that occurred in multiple isolates were explained by shared ancestry.

The simulation analysis described above found that the most targeted genes included a putative *S*-adenosyl methionine-dependent methyltransferase (A_02623) that has not been studied

in *Bordetella* sp. (ten different mutations, unadjusted $p < 10^{-5}$) and the transmembrane histidine kinase sensor protein, BvgS (11 different mutations, unadjusted $p = 0.01$). In *B. pertussis*, BvgS has been studied extensively as a master regulator of virulence[37], though its role in *B. hinzii* is unknown. The related RisS homolog (annotated as EnvZ in *B. hinzii*) that contains a substitution common to all of the DnaQ E9G lineages was also frequently mutated (eight SNVs in different lineages, unadjusted $p = 0.006$).

A gene ontology analysis performed on the set of frequently mutated genes identified in the simulation study above demonstrated significant enrichment (Fisher's exact test, $p < 0.05$) of genes involved in dicarboxylic acid metabolism, gluconeogenesis, and long-chain fatty acid metabolism (Supplementary Fig. 11). The mutational targets involved in these two pathways included genes putatively annotated as the tricarboxylic acid cycle enzymes fumarase (*fumC*), and malate hydrogenase (*maeB_1*), and the gluconeogenesis enzymes triosephosphate isomerase (*tpiA*), and fructose 1,6-bisphosphatase (*fbp*). Other individual targets were distributed among functional categories including glutamine synthesis and transport, and lipopolysaccharide synthesis and export. As a control, we also performed simulations using the mutational data from the in vitro MA experiments. This analysis did not find evidence of significant enrichment in the above targets during in vitro passaging, arguing that they are not simply mutational hotspots (Supplementary Data 2).

Gene deletion and the creation of pseudogenes have accompanied human adaptation in other *Bordetella* species[7,8]. In addition to the amino acid substitutions discussed above, 201 potential pseudogenes were created by indels and stop codons among the isolates (Supplementary Data 4). A gene ontology analysis demonstrated pseudogene enrichment in pathways relating to potassium ion transmembrane transport, proteolysis, cell adhesion, response to heat, and regulation of nitrogen compounds (Supplementary Fig. 12). Additionally, a gene annotated as thioredoxin reductase was both one of the most highly mutated genes in the isolate set and also a pseudogene in one lineage. These findings suggest that modification of the

activity of the thioredoxin system, which funnels NADPH to reduce disulfide bonds in the cell, might have conferred a selective advantage under the conditions encountered in vivo[38].

**Aggregation binds together genetically diverse lineages in single CFUs.** An unusual observation noted above was that a number of single colonies isolated from primary cultures contained multiple genetically distinct morphologies (Figs. 1, 2). This suggested that the underlying CFUs were composed of multicellular aggregates of distinct lineages. Supporting the hypothesis of aggregation, some of the isolates grown in broth culture demonstrated significant macroscopic clumping (Supplementary Fig. 13). Autoagglutination has been documented and studied among *Bordetella* species, though the exact mechanisms mediating aggregation in *B. hinzii* have not been well studied[39,40].

As the hypermutators would be expected to generate additional genetic diversity during the 2–3 passages required to separate the aggregated morphologies in vitro, we estimated the potential contribution of this component to the total diversity observed in the separated isolates (Supplementary Table 6). This analysis demonstrated that for some of the more closely related isolates, a sizeable component of the genetic separation could be attributed to in vitro passage, indicating that morphologically distinct isolates in the mixed-morphotype colonies were in fact even more genetically similar than implied by the sequences of the separated isolates. This was not true for the more distantly related isolates, but this finding nevertheless suggests a remarkable degree of phenotypic plasticity relative to the degree of genetic change.

## Discussion

In this work, we studied the long-term adaptation of the emerging zoonotic pathogen *B. hinzii* in an immunodeficient human host. Genomic analysis of serial isolates collected from both blood and gastrointestinal cultures over the course of 45 months revealed a striking degree of genetic diversification and provided a detailed view of the molecular mechanisms underlying within-host evolution. We found that the primary factor driving genetic diversification was a proofreading deficiency due to an inactivating mutation in the active site of the ε-subunit of DNA Polymerase III (DnaQ). Though isolates with an intact DnaQ gene were cultured from stool in the first culture (day zero) and from blood in the second culture (day 139), the DnaQ E9G isolates overtook the population by the third culture (day 244). Elevated mutation rates in these isolates generated secondary compound hypermutators with mutations in *mutM* and *mutY* genes, resulting in further increases in spontaneous mutation rates and alterations in spectra. Together these hypermutators persisted for at least 12 months, during which time no wild-type DnaQ isolates were cultured. Following this period, cultures revealed the apparent extinction of the entire hypermutator population, and periodic sampling revealed negative clinical blood and stool cultures for a period of 16 months. A *B. hinzii* lineage with wild-type DnaQ then reemerged two additional times (days 1014 and 1370) in blood cultures a year apart. Remarkably, these late isolates were very close to the LCA, suggesting the long-term persistence of a founder clade in the patient.

Hypermutators have been shown to occur in chronic infections and also on the shorter time scale of acute infection[2–6]. However, the importance of hypermutation in the context of zoonotic infection in a non-native host has not previously been studied in detail. In the case presented here, our findings argue that hypermutation resulted in a selective advantage through secondary adaptive mutations that were discovered and selected during the early course of clinical infection. These conclusions are independently supported by the observation that the DnaQ E9G

hypermutators overtook the wild-type founder lineage in both blood and gastrointestinal compartments for a period of at least 12 months with clonal succession by multiple divergent hypermutating lineages, suggesting the superior in vivo fitness of these lineages. Several genes were repeatedly mutated in DnaQ E9G isolates and had an excess of non-synonymous mutations with dN/dS ratios >1. This suggests both that these genes were under positive selection and that the elevated mutation rates facilitated the discovery of adaptive mutations in DnaQ E9G. Though the hypermutator population dominated clinical cultures for a prolonged period during the clinical infection, the DnaQ E9G population eventually disappeared, and it is possible that this was a consequence of reduced viability resulting from the cumulative mutational load.

It is well-known that immunodeficiency syndromes, both genetic and acquired, can confer highly specific susceptibilities to infection[41]. In these cases, the nature of the infection vulnerability phenotype reflects the physiology of the immune lesion, creating a biological niche permissive to invasion by specific organisms. Examples include nontuberculous mycobacterial infection in GATA2 deficiency[42], *Staphylococcus aureus* and *Burkholderia* infections in chronic granulomatous disease[43], disseminated coccidioidomycosis in STAT3, STAT1, IFN-γR, and IL-12-Rβ deficiency and dysregulation syndromes[44], and cryptococcal infection in idiopathic CD4+ lymphopenia[45]. In such cases, the pathogens exploit specific deficiencies in host defenses, and the dynamics of this exploitation reciprocally define the fitness landscape guiding the within-host evolution of the pathogen. It may thus be expected that idiosyncratic impairments in host defenses will leave unique imprints in the bacterial genome under selection in the context of chronic infection and that the detailed study of microbial evolution may reveal details about the nature of the host immune deficiency and the host–pathogen relationship that might not otherwise be obvious.

A notable genomic finding in the case studied here was a G:C > T:A transversion excess, which can result from oxidative lesions that generate 8-oxoguanine and other derivatives. In this host with impaired IFN-γ dependent oxidative defenses, we hypothesize that the observed oxidative mutational signature was a consequence of persistent exposure to prolonged, but sublethal, oxidative attack. Such a pattern of sublethal oxidative mutagenesis may indeed be a more general biological signature of impaired host oxidative defenses that provide low-level, but not sterilizing, oxidative immunity. It is furthermore tempting to speculate that from an evolutionary perspective, such nonlethal oxidative stress may have in fact facilitated adaptation by providing genomic diversity on which selection could act. The preservation of G:C > T:A mutations in the identified genomic targets of selection provides support that a subset of these mutations was in fact adaptive.

Detailed analysis of the targets of mutation suggested adaptation in at least three principle functional categories: intermediary metabolism, cellular redox maintenance, and signaling through two-component histidine kinase systems. The observed targets of intermediary metabolism included a number of functionally interrelated enzymes of the TCA cycle and gluconeogenesis, suggesting the possibility of specialized metabolic adaptation to the conditions within the host. The TCA cycle targets included mutations in the putative isocitrate dehydrogenase (*icd_1*) and phosphoenolpyruvate regulatory protein (*ppsR*) genes, both common to all of the DnaQ E9G isolates, and repeated mutations in the putative fumarase (*fumC*), and malate hydrogenase (*maeB_1*) genes. The gluconeogenesis targets included triose-phosphate isomerase (*tpiA*) and fructose 1,6-bisphosphatase (*fbp*). While metabolic studies will be necessary to define the consequences of these mutations, it is interesting to note that the

human-adapted relative, *B. pertussis*, evolved an unusual metabolic specialization in that it relies on TCA cycle catabolism of glutamate and related substrates as main carbon sources rather than glycolytic catabolism of sugars, with consequent dependence on gluconeogenesis for the supply of anabolic sugar precursors[7,46,47].

Glutathione and thioredoxin/thioredoxin reductase are critical components for maintaining cellular redox homeostasis and defense against oxidative stress. Targets involved in glutathione handling underwent mutation or inactivation during host adaptation, including genes annotated as a putative glutathione *S*-transferase (*gstB_5*), and glutathione hydrolase (*ggt*), among others. The linked thioredoxin/thioredoxin reductase system participates with glutathione in redox homeostasis and defense against oxidative attack by transferring reducing equivalents from NADPH to disulfide bonds in proteins. Remarkably, a gene annotated as a putative thioredoxin reductase (*trxB_1*) was both one of the most highly mutated genes with seven variants across the isolate set and also a pseudogene in one lineage. Together these findings in the glutathione and thioredoxin systems suggest specialized adaptation in the systems responding to oxidative damage, likely to be present from the mutational spectra observed. Whether a functioning thioredoxin system became maladaptive under the conditions of oxidative stress found in the host, perhaps by consuming excessive reduction potential, will need to be addressed by further experiments.

The third class of mutational targets included the *B. hinzii* homologs of the key histidine kinase transmembrane sensor proteins BvgS and RisS. The BvgS protein is a master regulator of virulence in *B. pertussis*, and signaling through this protein is required for *B. pertussis* colonization of the human respiratory tract[37]. RisS likewise plays a role as a master regulator of transcription in other *Bordetella* species, and additionally, the RisAS system interacts with the BvgAS system, though RisA may be the primary driver[34]. RisS is required for in vivo persistence in *B. bronchiseptica*[33], but appears to be a pseudogene in human-adapted *B. pertussis*[34]. The roles and potential interaction of the BvgAS and RisAS systems in *B. hinzii* are not well studied, and evaluation of the consequences of the observed mutations would require further analysis.

An unusual observation noted above was that a number of single colonies isolated from primary blood and stool cultures contained multiple morphologies that, when separated, were found to be genetically distinct. This suggested that the underlying CFUs were composed of multicellular aggregates containing genetically distinct members. A second related observation was that morphologically distinct isolates from the same mixed single colony were genetically more related to each other than to other isolates. This second observation could be explained by hypothesizing that diversifying lineages derived from a single cell remained physically bound together by aggregation through multiple generations. Furthermore, the lack of SNVs shared between colony aggregates from different days of collection may reflect regional isolation, similar to that which has been demonstrated in studies of *Pseudomonas aeruginosa* in cystic fibrosis respiratory disease[48]. This behavior may carry evolutionary consequences, as population aggregates containing greater genetic diversity than individual isolates may be more likely to contain surviving members under rapidly changing conditions as that found in a non-native host. Some of the isolates grown in broth culture demonstrated significant clumping, consistent with aggregation. Autoagglutination has been studied among *Bordetella* species[39,40], though the exact mechanisms mediating the aggregation seen here were not further explored in this work.

In conclusion, we have presented a genomic analysis of the events underlying the adaptive evolution of the emerging pathogen *B. hinzii* following presumptive zoonotic infection. We propose that early inactivation of DNA polymerase III ε-subunit proofreading activity was the primary driver of the observed genetic diversity and suggest a role for hypermutation in this case of post-host jump adaptation. More generally, our findings suggest a role for host immune phenotype in shaping within-host–pathogen evolution following zoonotic infection.

## Methods

**Bacterial isolates.** Clinical cultures were performed as part of routine diagnostic care for the patient. Deidentified bacterial isolates from these cultures were used for this work. Blood cultures were performed with an automated BD Bactec FX blood culture instrument (Becton Dickenson, Franklin Lakes, NJ). Stool cultures were plated to standard microbiologic culture media, including blood agar plates (Remel, Lenexa KS). Isolates from both sources were subcultured prior to identification, sequencing, and antimicrobial susceptibility testing. Initial identification was performed using a Bruker Microflex MALDI-TOF mass spectrometer (Bruker Daltonics, Billerica, MA) following the manufacturer's instructions. Serial isolation streaking was used to separate different morphologies that were mixed within single colonies. *B. hinzii* ATCC strains ATCC_51730, ATCC_51783, and ATCC_51784 were cultured from lyophilized stock on blood agar plates.

**Antimicrobial susceptibility testing.** Susceptibility testing of isolates was performed both at the National Institutes of Health and Clinical and Environmental Microbiology Branch, Centers for Disease Control. Testing at the NIH employed automated broth microdilution methods using a TREK Sensititre instrument (Thermo Fisher Scientific, Waltham MA) and custom TREK susceptibility panels. Testing at the CDC was performed by the broth microdilution method as described by the Clinical and Laboratory Standards Institute;[49] however, there are no interpretive breakpoints published by CLSI for *B. hinzii*. Reference frozen MIC panels were prepared in-house at the CDC. Cation-adjusted Mueller–Hinton broth (Difco, Becton Dickinson; Sparks, MD) was used for MIC testing at a final volume of 100 µL/well; MIC panels were incubated in ambient air at 35 °C for 24 h.

**Illumina WGS library preparation and sequencing.** Genomic DNA was extracted from 1 mL liquid culture using either the DNeasy Blood and Tissue kit (Qiagen, Valencia, CA) or Zymo Quick-DNA miniprep (Zymo, Irvine, CA), per the manufacturer's instructions. DNA yields were quantified with a Qubit 2.0 fluorometer (Invitrogen, Carlsbad, CA) and DNA quality was evaluated using gel electrophoresis with an 0.8% agarose gel (Lonza, Rockland, ME). DNA was fragmented using a Covaris M220 (Covaris Inc, Woburn, MA) to ~450 bp and whole-genome shotgun libraries were prepared using TruSeq Nano DNA kit with an Illumina NeoPrep system (Illumina, San Diego, CA), or the NEBNext Ultra II DNA Library Prep Kit for Illumina (New England Biolabs, Ipswich, MA) following kit instructions. Patient isolates were sequenced using an Illumina MiSeq in 2 × 300 bp mode with 600 cycle V3 sequencing kits to a depth of ~40x–120x (Supplementary Table 3). ATCC strains and selected patient isolates were additionally sequenced on an Illumina iSeq 100 instrument using libraries prepared with Illumina Nextera Flex kits per the manufacturer's instructions.

**PacBio sequencing.** Five isolates (1G1, 1G2, 2B1, 2B3, and 2G1) were sequenced with a PacBio RS II instrument (Pacific Biosystems, Menlo Park, CA). High-quality, high molecular-weight genomic DNA was isolated using the DNeasy UltraClean Microbial Kit (Qiagen, Germantown, MD, USA). SMRTbell libraries were then generated using the SMRTbell Template Prep Kit 1.0 (Pacific Biosystems, Menlo Park, CA) and size-selected using a BluePippin instrument (Sage Science, Beverly, MA, USA). SMRTbell libraries were primer annealed, Magbead, and polymerase bound using the MagBead Binding Kit v2 and DNA/Polymerase Binding Kit P6 v2 (Pacific Biosystems, Menlo Park, CA) and then sequenced using SMRTcells v3. Sequencing data were de novo assembled using HGAP 2.0 in the SMRT Analysis Portal (Pacific Biosciences, Menlo Park, CA). Overlapping contig ends were removed to circularize individual PacBio contigs, and short-read data was mapped to circularized contigs to detect/correct errors.

**Additional genomes used in this study.** Twelve complete *B. hinzii* genomes were downloaded from the NCBI database: *B. hinzii* F582, *B. hinzii* 621, *B. hinzii* H568, *B. hinzii* H720, *B. hinzii* NCTC13199, *B. hinzii* 14-3425, *B. hinzii* 243-2, *B. hinzii* 4134, *B. hinzii* 4449, *B. hinzii* NCTC13200, *B. hinzii* SV2, and *B. hinzii* TR-1212 (Supplementary Table 7).

**Genome assembly and initial phylogeny reconstruction.** Illumina reads were assembled with SPAdes v3.14.0[50] in careful mode using reads correction. Contigs that had a length of less than 500 bp were removed from the assemblies. A core genome was then constructed by comparing the genomes of the patient isolates with the set of NCBI *B. hinzii* genomes available publicly (Supplementary Table 7) using Roary v3.13.0[51] after annotation of all genomes with Prokka v1.14.6[52]. RAxML v8.2.12[53] was used to construct a rooted maximum likelihood tree using a

general time-reversible (GTR) substitution model with gamma correction for among site rate variation.

**Last common ancestor reconstruction**. Two methods producing equivalent results were used to reconstruct the LCA. In the first method, the PacBio genome assembly for isolate 2B3 containing a WT *dnaQ* gene was chosen as a reference for the initial reconstruction of the ancestral genome. Illumina short reads from the different patient isolates were aligned to the reference PacBio genome and variants were called using Snippy v4.4.0 (https://github.com/tseemann/snippy), with default options. To identify sequence variants specific to the patient isolates, we included all 12 *B. hinzii* genomes downloaded from NCBI as outgroups (Supplementary Table 7). Contigs in these assemblies were mapped to the 2B3 PacBio genome with snippy using the "-ctgs" option. For each variant in the set, the alternative allele was considered "ancestral" if it was found in the majority of the patient isolates, and the reference allele was not present in the majority of the outgroup genomes. The set of ancestral variants defined in this way was then integrated into the 2B3 genome using VCFtools v0.1.16[54] to create the genome of a likely LCA. This genome was then annotated using Prokka v1.14.6.

In the second method, the genomes of the patient isolates along with FDAARGOS_621 were extracted from the outputs of snippy variant calling against the 2B3 PacBio complete genome previously described. Those genomes were used to construct an aligned and concatenated core genome using Roary v3.13.0. From this, a rooted phylogenetic tree was constructed using RaxML v8.2.12 using a GTR substitution model with gamma correction for among site rate variation. The program phyloFit from the PHAST v1.4 package was then used first to fit the tree model to the multiple alignment of core genomes sequences by maximum likelihood, using the GTR substitution model. This tree model was used in conjunction with the sequences as input for PREQUEL from the PHAST package to reconstruct the most likely ancestral state at the node corresponding to the LCA (Supplementary Fig. 4). PREQUEL computes marginal probability distributions for bases at ancestral nodes in the phylogenetic tree. Finally, the reconstructed sequences of the two methods were compared. To perform genome-wide comparison we concatenated the CDSs from our original LCA into a similarly organized core genome, and both concatenated genomes were then compared using QUAST v5.0.2. In total 11 indel events were identified, all of which were located directly at, or less than 5 bp from the start or end of a gene. These indels were ignored as they likely correspond to artefacts created by the artificial gene junctions in the concatenated core genome.

**Variant calling relative to LCA and pseudogene identification**. Sequence variants were called using Snippy v4.4.0 with default parameters using the LCA as a reference. In cases in which hypermutator isolates were sequenced twice, only variants with support from both sequencing runs were retained to correct for additional mutations that occurred during passaging. Annotation of the variants was performed using SnpEff v5.0[55] and further processed using SnpSift and custom scripts. Medium and large deletions were identified with Breseq v0.35.1[56] and validated by visual inspection of aligned bam files. Pseudogenes were defined as genes containing nonsense mutations or frame-shifting indels that reduced the inferred length of the encoded protein by 10% or more. The genes encompassed by the two large deletions from isolates 8B1 and 1G1 were not considered as pseudogenes in this analysis.

**Phylogeny reconstruction**. Two phylogenies were reconstructed to account for the structure of the population, using variants from the DnaQ WT isolates relative to the DnaQ WT LCA (true ancestor) and DnaQ E9G variants relative to the DnaQ E9G clade founder genome, which was constructed by adding the 26 variants common to all DnaQ E9G isolates to the DnaQ WT LCA. For each set, the variants were called and the complete genome was reconstructed using Snippy v4.4.0. Those reconstructed complete genomes were annotated by Prokka v1.14.6 and the different ortholog coding regions from the different isolates were identified and aligned using Roary v3.13.0 along with each clade founder genome used as the respective reference. A maximum-likelihood phylogenetic tree was then constructed for each group with RAxML v8.2.12 using a GTR substitution model with gamma correction for among site rate variation. The phylogenies were then plotted using R v4.0.0 and the ape package v5.4 was used to join the two trees by their ancestor (Fig. 2).

**dN/dS calculations**. To compute the genome-wide dN/dS ratio, the number of synonymous sites and non-synonymous sites was first calculated for each gene using gene annotation information and sequences from the gff files. In order to evaluate the difference in proportions of non-synonymous substitutions, we aggregated data either for the whole genome or for the 24 genes containing mutations in the DnaQ E9G clade founder by summing the number of synonymous and non-synonymous sites and the number of synonymous or non-synonymous mutations. The genome-wide dS rate was used as a denominator in the gene-by-gene calculations displayed in Supplementary Data 3, as many individual genes had no synonymous substitutions.

**Homoplasy test**. To look for homoplasy, the R package homoplasyfinder v 0.0.0.9000 was used[57]. A tree was constructed in RaxML v8.2.12 by using the concatenated core genome of all the patient isolates. The tree was then compared to a table containing the information of the presence and absence of the variants for the different patient isolates. Variants with a consistency index below 1 were identified as variants conflicting with the topology of the tree, which could be explained by more than one mutational event or recombination. A total of 85 out of 6459 mutations were identified as homoplastic by this analysis.

**Simulation**. In order to identify potential targets of selection, we looked for genes in which the number of observed substitutions was significantly higher than expected by chance. The initial homoplasy calculation above indicated that only 1.3% of mutations in different isolates were homoplastic. Thus, the simulation that follows includes only the nonredundant set of unique mutations; that is, we include in the analysis almost exclusively independent mutational events. A simulated dataset was then created in which this set of nonredundant unique substitutions observed in the patient isolates were randomly redistributed across the totality of the genome 100,000 times. Because of the potential bias introduced by the differences in the number of substitutions per isolate based on their degree of divergence from the ancestor, the isolates were separated into three groups: (1) Low divergence: DnaQ WT, (2) Intermediate divergence: DnaQ E9G, and (3) High divergence: DnaQ E9G with secondary *mutY* and *mutM* mutations. The total number of intragenic and intergenic substitutions were counted in the different groups. The mutations that were shared among groups, including the 25 substitutions shared by all the DnaQ E9G lineage were taken into account in order to avoid overcounting. For each iteration of the simulation, the mutations were redistributed in the reconstructed ancestral genome. After 100,000 cycles of redistribution, the results of the different groups were analyzed independently and then merged. With the distributions of numbers of substitutions per gene, we computed an unadjusted empirical *p* value for each gene. This value indicates the unadjusted probability of observing a number of substitutions equal to or greater than the number observed in the patient isolates by chance by using the complement of the empirical cumulative distribution function, ecdf, adapted from the standard R function:

$$P_{\text{empirical}}(t) = 1 - ecdf(t-1) = 1 - \frac{\text{number of elements in the sample} \le (t-1)}{n}$$
$$= 1 - \left( \frac{1}{n} \sum_{i=1}^{n} I_{xi \le (t-1)} \right)$$

In the context of the simulation, for one gene, after *n* cycles of the simulation x = (x1, x2,… xn), $P_{\text{empirical}}$ (t) is the fraction of the total cycles where the same number *t* or more than what was observed in the patient was found. A *z*-score was also calculated to quantify the difference in the number of mutations actually found in each gene in the patient isolates from the expectation values set by the randomly distributed variants of the simulation. In order to compare the effects of selection in the patient to an independent "neutral drift" dataset, the same simulation approach was used on the in vitro MA data. Because of the low mutation count observed in the WT group during the longitudinal MA experiment, only the parallel MA results for the DnaQ E9G mutants and the DnaQ E9G MutY and MutM double mutants were redistributed. The simulation was run for 100,000 cycles as previously, the results for both groups were merged, and the empirical *p* value and *z*-scores were calculated. The results of both approaches were then compared.

**Annotation and gene ontology analysis**. To perform functional annotation, the complete set of the ancestor protein sequences was compared to the NCBI nr database using BLAST v2.10.0[58] followed by GO annotation using blast2go v5.2.5[59] and InterProScan v5.42-78.0[60]. Different GO terms annotated by these approaches were subsequently merged prior to downstream analysis. Gene set enrichment analysis was performed with topGO v2.38.1. Across the 3900 GO annotated genes, 2759 were suitable for analysis in topGO and used with the weight01 algorithm in order to perform Fisher tests. Individual gene sets were selected for gene set enrichment analysis as described in the Results.

**Overview of mutation accumulation experiments**. To evaluate spontaneous mutation rates and spectra in the absence of selection, we performed MA experiments. Individual isolates were serially passaged on blood agar plates followed by whole-genome sequencing. DnaQ WT isolates with low spontaneous mutation rates were passaged 30 times per MA line (~3070 generations in aggregate after discounting shared passages, see details in the following sections). To mitigate the effect of high mutation rates in DnaQ E9G isolates and to reduce the impact of secondary mutations, MA experiments were performed in parallel and using only one passage. Following DNA extraction from 1 mL of liquid culture, DnaQ WT isolates were sequenced and analyzed as described above. For DnaQ E9G isolates, sequencing libraries were prepared in 96-sample batches using the high-throughput iGenomX Riptide library prep kit (iGenomX, Carlsbad, CA) followed by sequencing on an Illumina NextSeq 550 (see following sections for details). Initial variant calling was performed with Snippy v4.4.0 using the DnaQ E9G LCA as a reference and with a minimum of threefold coverage. For

downstream mutational spectra analysis, we retained only variants that were unique within the entire DnaQ E9G MA dataset (see following sections for details).

**Longitudinal mutation accumulation experiments with patient DnaQ WT isolates and ATCC strains.** For the first longitudinal MA experiment, the two DnaQ WT isolates 2B3 and 7B1 and three reference ATCC strains ATCC_51730, ATCC_51783, and ATCC_51784 were included. Isolates from these five strains were plated from the frozen stock onto blood agar (Remel, Lenexa KS) and incubated at 35 °C for 3 days. For each isolate, two 1 mm colonies were picked and replated on two separate blood agar plates creating two independent lineages for each of the five strains (ten total lineages). Isolates were then passaged every 2 days by picking a 1 mm colony and streaking it on a new blood agar plate. At the second passaging step, two 1 mm colonies were picked and replated on two independent blood agar plates creating four independent lineages for each strain (20 total lineages). After 29 passaging steps, DNA was extracted from a 1 mL liquid culture prepared using one colony from each of the 20 independent lineages using Zymo Quick-DNA miniprep (Zymo, Irvin, CA), and DNA yield and quality were checked. DNA was fragmented using a Covaris m220 (Covaris Inc, Woburn, MA) to ~450 bp and sequencing libraries were prepared using the NEBNext Ultra II DNA Library Prep Kit for Illumina (New England Biolab, Ipswich, MA) following the kit's instructions.

In parallel with the above experiment, the number of cells and generations corresponding to a 1 mm colony were estimated. For each of the DnaQ WT patient isolates and ATCC strains, a 1 mm colony from one plate was suspended in 1 mL PBS followed by plating dilutions on blood agar plates in duplicate. Colonies were counted after 2 days of incubation at 35 °C, and the average number of cells per 1 mm colony was estimated assuming that each colony was derived from one colony forming unit (average of two duplicate plates). For both the wild-type and ATCC strains, the number of cells in a colony ranged from $4.05 \times 10^7$ to $5.30 \times 10^8$, with an average of $1.72 \times 10^8$, corresponding to an average of 27.2 generations by passages based on the log2 of the cell count.

**Variant calling approach in the longitudinal mutation accumulation experiments.** The variants were called using Snippy with the same options as previously, except that the min cov was set to 3. Isolates 2B3 and 7B1 were compared to the consensus sequence reconstructed by Snippy from the ancestor by applying the identified variants from the patient isolates. ATCC_51730 was compared to the F582 genome, a clone of ATCC_51730 for which a complete circularized genome was available from the NCBI dataset. Lastly, ATCC_51783 and ATCC_51784 were compared to their respective assemblies previously made using spades v3.14.0. Each potential variant was manually inspected and variants for which evidence was found in at least three of the four lineages from the same founding strain were removed in order to keep only independent mutations that appeared during the passaging steps of the experiment.

**Parallel mutation accumulation experiments with the DnaQ E9G mutants.** Mutation rates and spectra for the 18 DnaQ E9G isolates were assessed in a second MA experiment. Because of the higher expected mutation rate, the number of total passages per isolate was minimized in this experiment to reduce the total mutational load on each genome, which might independently modify mutation rate through mutations in DNA repair or other processes. This was accomplished through a highly parallel experimental design. However, it is fully expected that even with this approach, secondary modifiers of mutation rate and spectrum will arise within the population, given the very large number of mutations. To account for the variability of growth rate, we separated the isolates into three groups that would be passaged after an incubation of either 2, 3, or 5 days as required to reach 1 mm colony size on solid media. We first plated the isolates from frozen stock on a blood agar plate, and after the corresponding incubation time, a 1 mm colony for each isolate was resuspended in PBS. The solution was diluted by a factor of $10^5$, and 100 μL was then used to make 4 to 39 independent blood agar subcultures. The isolates were then incubated at 35 °C and 1 mm colonies were picked from the plate and grown in 200 mL LB in a 96 well plate at 35 °C under agitation in a Cytation 3 (BioTek, Winooski, VT) for 2 to 5 days. The solution was then pelleted, and DNA extracted using DNeasy 96 Blood & Tissue Kit (Qiagen, Valencia, CA), following the manufacturer's instructions. The yield of the extraction was quantified using a Qubit 2.0 fluorometer. Some lineages that yielded low DNA concentrations were sequenced in replicate, and only the replicate with the best yield was used in this analysis. Libraries were prepared using the iGenomX RIPTIDE High Throughput Rapid Library Prep Kit (iGenomX, Carlsbad, CA) following the protocol for high GC content.

**Variant calling approach in the parallel mutation accumulation experiments.** In this approach, the reads for each lineage were aligned to the genome of the reconstructed ancestor using BWA 0.7.17 and soft clipping was allowed for a maximum of 10 bases using samclip v0.2. Duplicates were then removed and the. bam file was sorted using SAMtools v1.9. Variants were called using FreeBayes v1.3.1 in haploid mode and only accepting variants with a base quality of at least 30 and a mapping quality of at least 60. The variants were then filtered using BCFtools to keep calls with a QUAL of at least 100, a minimum coverage of 3, and a minimum alternative allele frequency of 0.66. The variants were then passed to Vt

Normalize v0.5 to get consistent variant callings across the lineages. Finally, the ancestral variants present at the start of the experiment were removed to keep only those mutations that occurred during the experiment. Next, genomeCoverageBed from the BEDtools v2.72.1 package was used to identify which regions of the genome were at least covered by three reads in all the lineages, in order to compare the same regions of the genome across the different lineages and isolates. We then looked at the remaining variants that were common between different lineages. We were able to show that the majority of variants were shared by isolates derived from the same founding isolates, which was expected due to their relation in the first passage from the frozen stock. However, eight mutations were also identified that were shared across lineages descending from different founders and these were identified as true variants confirmed by inspection of mapping reads. To retain only the mutations that appeared independently during the last passage, we removed the mutations that appeared more than once across lineages in order to exclude the mutations that occurred during the first passage from frozen stock and before separation.

Independent mutations for each of the patient isolates were then merged together using vcf-merge, and the mutational spectra from the lineages were extracted. In order to eliminate the potential impact of variable local genomic composition on mutational spectra, we made efforts to ensure that the same regions of the genomes were compared between isolates in cases where there were coverage differences. To do this, we first masked all the variants that were in regions that were not covered by at least three reads in the parallel MA dataset, in order to compare regions with similar compositions. The mutational spectra from the patient isolates were then extracted using an in-house script. The double panel heatmap in Fig. 4 was constructed using the R package ComplexHeatmap v2.2.0. The 95th percentile confidence intervals in Supplementary Fig. 5 were calculated with the exact binomial confidence interval function in the R package.

**Reporting summary.** Further information on research design is available in the Nature Research Reporting Summary linked to this article.

**Ethical Declaration.** Informed written consent was obtained from the patient under NIH IRB protocol 93-I-0119 upon admission to the NIH Clinical Center, approved by the NIH IRB committee. Diagnostic clinical cultures were performed as part of routine standard-of-care management under this consented protocol, and only de-identified subcultured bacterial isolates were used in the work presented in this manuscript. The results of the work in this manuscript were not used for patient care and did not form the basis of clinical intervention. The work presented in this manuscript was thereby excluded from further NIH IRB review, on the basis of the fact that it was a study of a single case, involving only sequencing and analysis of bacterial isolates.

## Data availability

Sequencing data have been deposited in the NCBI Sequence Read Archive under the accession number PRJNA625574. Primary data have been deposited with Zenodo: https://doi.org/10.5281/zenodo.4929939. Requests for bacterial isolates from this study are subject to a negotiated Material Transfer Agreement with the NIH and US Government. Source data are provided with this paper.

## Code availability

All custom scrips used for calculations have been deposited with Zenodo: https://doi.org/10.5281/zenodo.4894884.

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

## Acknowledgements

This work was supported in part by the Intramural Research Program of the National Institute of Allergy and Infectious Disease. We thank Patrick Mc Gann and the late Erik Snesrud (Walter Reed Army Institute of Research, Silver Spring, MD) for providing PacBio sequencing of selected isolates; Steven Holland for invaluable advice and feedback; Cindy Palmer and Beatriz Marciano of the clinical team; the laboratory staff from the Division of Healthcare Quality Promotion at the Centers for Disease Control and Prevention, Atlanta, GA for the performance of susceptibility testing; and the staff of the NIH Microbiology Service, Department of Laboratory Medicine, NIH Clinical Center for assistance with isolate cultures. This work utilized the computational resources of the NIH HPC Biowulf cluster. (http://hpc.nih.gov). The content of this publication is solely the responsibility of the authors and does not necessarily reflect the official views or policies of the Department of Health and Human Services, National Institutes of Health, Centers for Disease Control and Prevention, or the Department of Defense, nor does mention of trade names, commercial products, or organizations imply endorsement by the US Government.

## Author contributions

A.L., P.P.K., and J.P.D. conceived of and designed the study. A.L., C.J.W., A.D.C., J.H.Y., and P.P.K. performed experiments and genomic sequencing. A.L., C.J.W., P.P.K., and J.P.D. performed all data analysis. A.L., P.P.K., and J.P.D. generated figures and wrote the manuscript. All authors critically reviewed and edited the manuscript.

## Funding

## Competing interests

The authors declare no competing interests.
