## [Peer Review File · Nature Communications]

Reviewers' Comments:

Reviewer #1:

Remarks to the Author:

This is a very interesting manuscript describing the adaptation of an unusual pathogen, *Bordetella hinzii*, in a chronic infection of an immune-deficient patient. A small number of similar studies have been described, but not in this group of organisms, and this is a valuable addition to the literature. The analysis of mutational spectra into hypermutation and oxidative damage using experimental mutational accumulation is a very interesting and novel aspect of the work.

In general, the work seems sound and well described. While it is certain that studies of this type do give an insight into host adaptation, it is not at all clear that they form an actual part of historical host-jumps. I therefore think that the claim in the abstract and discussion that "These findings suggest a fundamental role for host immune phenotype in shaping pathogen evolution following a host-jump." is somewhat overstated. The combination of a hypermutating bacterium and an immune-deficient host certainly sheds light on the longer-scale process of adaptation, but is not in itself evidence that immune-deficient hosts are actually involved in host-jumps. For that the authors would need to show the signatures of this effect in other host-adapted *Bordetella* that had made the jump, to be sure this was a general pathway. Also, a host-jump needs onward transmission. There does not seem to be any evidence this adapted bacterium is competent for that, and indeed the lineage subsequently died out even within this one patient. These statements should be more carefully qualified.

I have a couple of main concerns about the analysis and interpretation of the data. First, the authors appear have used unsophisticated and ad-hoc phylogenetic reconstruction methods, although the text, figures and methods are unclear on this. They should really repeat the analysis with maximum likelihood or Bayesian approaches, which will allow them to perform a more accurate analysis of the data. Second, some of the discussion about selection is confused and again based on simplistic analysis. Details of these are below.

Figure 2: NJ tree. Why not use ML? It is much more robust, and would allow dating – is there a temporal signal in the tree? Also, why is the tree not rooted? It should be easy to find an outgroup. It says in the methods that RaxML was used, but only an NJ tree is shown.

Line 115: "An additional apparent deletion of a 38 Kb phage-like element in isolate 8B1 may alternatively represent insertion of this element into an ancestor shared by all isolates except 8B1." These alternatives can be easily discriminated if the tree is properly rooted.

And again in lines 119-123 and Suppl Methods. They seem to have gone for an ad-hoc method of recreating an LCA, rather than using a robust phylogenetic approach with a rooted tree.

The discussion of "two ancestors" from line 126 is also confusing. If this is assumed to be a single infection there will only be one ancestor of the whole infection, but the high-divergent strains will share a LCA within that tree.

Lines 131 to 139. The fact that isolates from mixed morphology colonies are more similar to each other than isolates taken on different days could be explained by there being geographically separate foci of infection that are evolving independently. This behaviour has been reported using deep sequencing in chronic lung infections such as *Pseudomonas aeruginosa*. This should be discussed.

Line 175. I would disagree that hypermutation itself has a selectable advantage. It is certainly the case that variants with a selective advantage arise more rapidly on a hypermutator background, but the hypermutation variant does not itself confer a selective advantage. Selection does not have foresight. A variant cannot be selected because it will have an advantage at a later date.

The hypermutator variant increases in the population because it hitchhikes on the selective variants that arise later.

This also means that these variants are not "co-selected" with the hypermutation variant (Line 177). They are the primary selected variants, and the hypermutation variant is a hitchhiker.

This is expanded in the discussion (line 308), with the concept that additional hypermutator lesions enhance "adaptive evolutionary searches". Again, selection has no foresight. A mutation cannot be selected because it will provide a benefit in the future (after the search). It may appear to be the case after the fact, but that is only because it has hitchhiked to higher frequency on the back of selective mutations that have occurred on that genetic background. This may appear to be a semantic argument, but I think it is important to be clear about selective processes.

Line 177. Positive selection is not indicated by the number of non-synonymous changes. Mutations arise randomly, and non-synonymous variants arise more frequently than synonymous ones because they have more opportunity to do so (there are more sites in the genome where non-synonymous mutations can occur). To show positive selection, the authors must calculate: (number of non-synonymous variants per non-synonymous site) / (number of synonymous variants per synonymous site); i.e. dN/dS; and then show that it is significantly different from the neutral expectation.

The section beginning line 220 is confusing. The authors say they are looking for genes which were "independently mutated in different patent isolates". However, the simulation described in the supplementary material is a permutation test looking at mutation burden in each gene. It does not test whether those mutations occurred independently in different isolates (i.e. convergent evolution). For that, they would need to do a homoplasy test using a phylogenetic tree, which they do not describe. They should change the description or do a homoplasy test (or both).

Minor points:

Line 227. Please correct "mutated x times"

Reviewer #2:

Remarks to the Author:

Launay A et al. described in vitro characterisation of a emergent pathogen, *Bordetella hinizzi*, found in a patient with an inborn error of immunity (IEI). Following a period of 45 months and using a clonal lineage, the authors detected a diversification of genetic in the bacteria. The description of microbiological procedures is very well documented. However the link between the microbe and the host is not clear. Several data concerning IL-12R β 1 deficient patient are missing.

B. hinizzi can be a cause of infectious disease in human (endocarditis, bacterimia, pneumonia ...). Why the authors followed the diversity of phenotype in two samples : blood and stool ? they had access to other samples ? *B. hinizzi* infection was localized or disseminated ?

What is the infectious history in the IL-12R β 1 patient ? it includes other pathogens found in IL-12R β 1 deficient patients, like salmonellosis ? what is the mutation ? In general this IEI has an incomplete penetrance of mycobacteria and salmonellosis. I am curious to know if other members of the family carried the mutation in the same state of patient and if the individuals ave also a diversity of genetic in *B. hinizzi*.

In the discussion, the authors claim that some IEI have selective predisposition to some microbes. I agree with this comment. However, the selection of IEI is not clear, in particular idiopathic CD4 lymphopenia which by definition none genetic mutations were found. What is the connection around the cited IEI ?

Minor comments

Patients with IL-12Rb1 deficiency have a reduced levels of IL-12/IL-23 stimulated IFN- γ production. The stimulation with both cytokines is affected (second paragraph of introduction).

Response to Reviewers

In Vivo Evolution of an Emerging Zoonotic Bacterial Pathogen in an Immunocompromised Human Host

Launay A, Wu CJ, Dulanto Chiang A, Youn JH, Khil, PP, Dekker, JP

Manuscript # NCOMMS-20-41592-T

Reviewer #1 (Remarks to the Author):

This is a very interesting manuscript describing the adaptation of an unusual pathogen, Bordetella hinzii, in a chronic infection of an immune-deficient patient. A small number of similar studies have been described, but not in this group of organisms, and this is a valuable addition to the literature. The analysis of mutational spectra into hypermutation and oxidative damage using experimental mutational accumulation is a very interesting and novel aspect of the work.

Response: We thank the reviewer for the careful reading of the manuscript and positive evaluation of our work. Below we respond to each of the reviewer's questions and suggestions. We have substantially revised the manuscript and included additional analyses as requested. We believe the revised manuscript is greatly improved and we thank the reviewer for the insightful comments and assistance. Please note that an additional section explaining the methods used in the responses to this reviewer is included at the end of the RTR. Note that all line numbering below refers to the tracked version of the manuscript for more rapid identification of the changes.

In general, the work seems sound and well described. While it is certain that studies of this type do give an insight into host adaptation, it is not at all clear that they form an actual part of historical host-jumps. I therefore think that the claim in the abstract and discussion that "These findings suggest a fundamental role for host immune phenotype in shaping pathogen evolution following a host-jump." is somewhat overstated. The combination of a hypermutating bacterium and an immune-deficient host certainly sheds light on the longer-scale process of adaptation, but is not in itself evidence that immune-deficient hosts are actually involved in host-jumps. For that the authors would need to show the signatures of this effect in other host-adapted Bordetella that had made the jump, to be sure this was a general pathway. Also, a host-jump needs onward transmission. There does not seem to be any evidence this adapted bacterium is competent for that, and indeed the lineage subsequently died out even within this one patient. These statements should be more carefully qualified.

Response: We thank the reviewer for this feedback and we agree that our speculations regarding the role that immune-deficient hosts might play in actual historical host jumps were overly broad. We also agree that we have not presented sufficient evidence of competence of the host-adapted isolates for onward transmission. We have thus modified the text in lines 45-46, 89-90, and 616-619 to clarify and more carefully qualify these speculations.

I have a couple of main concerns about the analysis and interpretation of the data. First, the authors appear have used unsophisticated and ad-hoc phylogenetic reconstruction methods, although the text, figures and methods are unclear on this. They should really repeat the analysis with maximum likelihood or Bayesian approaches, which will allow them do perform a more accurate analysis of the data. Second, some of the discussion about selection is confused and again based on simplistic analysis. Details of these are below.

Response: We appreciate the reviewer's concerns regarding the methods we used and data interpretation. We tried to make the analysis accessible to broadest audience possible, and in a few cases used less sophisticated but easier to understand approaches when we believed it did not affect the conclusions. In addition, as reviewer also noted, the description of the approaches that we used was sometimes not sufficiently clear. We have clarified description of the methods in appropriate places, and have rewritten parts of the results and discussion sections to improve technical rigor and clarity as described below. In addition, we repeated several analyses using maximum likelihood approaches and found essentially identical results as described. We respond to the specific details below.

Figure 2: NJ tree. Why not use ML? It is much more robust, and would allow dating – is there a temporal signal in the tree? Also, why is the tree not rooted? It should be easy to find an outgroup. It says in the methods that RaxML was used, but only an NJ tree is shown.

Response: First, we thank the reviewer for identifying an error in the figure legend text. The tree that is represented in the figure was in fact calculated using a maximum likelihood approach with RaxML. The reference to an NJ tree was from a prior version of the manuscript that was not properly edited. We have corrected this error in the figure text. Regarding the choice to use an unrooted tree, this was mainly aesthetic as we believe this allows for a clearer visualization of the star-like appearance of the phylogeny. However, in response to the reviewer's comment, we have additionally presented a rooted tree using *B. hinzii* FDAARGOS_621 as an outgroup (Fig R1 below, detailed approach described in Methods section at end). Comparing the structure of the rooted tree to the unrooted tree from the manuscript (Fig 2), it is clear the topology is essentially identical. We have added the rooted tree to the manuscript as Figure S4 and included the approach to its construction in the methods section.

Figure R1: Maximum likelihood tree constructed using the core genome in all patient isolates. *B. hinzii* with strain FDAARGOS_621 used as an outgroup. A cladogram is shown to better illustrate the topology. The numbers at the node of the tree indicate bootstrap support, and the red dot indicate the node used for LCA reconstruction.

While the trees displayed in Figure 2 and Figure R1 above contain little obvious evidence of temporal signals, and the topology is essentially star-like, we agree with the reviewer that analysis of temporal signals in the data could add valuable insights. To further search for evidence of temporal signal, we performed deeper analysis of the SNVs. In Figure R2, it is apparent that with the exception of 25 SNVs found in the DnaQ E9G clade founder, relatively few SNVs are shared between isolates, and there is a substantial heterogeneity in SNV divergence from the LCA driven by hypermutator phenotypes.

Figure R2. Shared SNVs in patient isolates. All of the isolates are divided into two groups – WT DnaQ and DnaQ E9G – and then ordered by date of collection within each group. The entries represent the numbers of shared SNVs compared to the LCA.

We thus separated all of the isolates into three broad groups based on defects in DNA repair mechanisms (and resulting divergence in mutation rates) and plotted the divergence from LCA versus collection time (Figure R3). Taken together, there appears to be an increase in divergence from the LCA over time for both hypermutator isolates and wild-type. However, the numbers of isolates are low and although trends are qualitatively apparent, there are some obvious outliers. Since hypermutator isolates were likely derived from the LCA, we focused on the isolates with wild-type DnaQ.

Figure R3. Divergence of PT isolates from LCA versus collection date.

A plot of divergence from the LCA versus time for the wild type DnaQ isolates only (Figure R4), demonstrates the same trend: the later isolates have diverged further from the LCA. A linear regression best fit gives a rate of ~ 7 SNV/year, which is a reasonable estimate compared to similar values for bacterial intra-host adaptation. Still, since the study contains only 4 wild-type isolates, the linear fit is not great (R-squared: 0.729), and divergence is substantial - 95% CI interval for slope is very wide: -0.5 – 18.3 SNV/year. Additionally, we understand that we use the simplest possible evolutionary assumption of constant divergence rate, but the small number of isolates does not allow fitting of more complicated models. In summary, although there are clear indications of temporal divergence in our isolates, we believe the dataset is too small and our statistical power is too limited to allow us to make any strong conclusions. We thus chose to leave such analysis out of the original version of the manuscript. We now have included additional discussion in the text in lines 163-165, and added Figures R2 and R4 in the supplement of the revised manuscript as Figures S5 and S6.

Figure R4. Divergence of PT isolates with wild-type DnaQ gene from LCA versus collection date. Also shown line of linear regression fit (+/- 95% CI).

Line 115: "An additional apparent deletion of a 38 Kb phage-like element in isolate 8B1 may alternatively represent insertion of this element into an ancestor shared by all isolates except 8B1." These alternatives can be easily discriminated if the tree is properly rooted.

Response: We agree with the reviewer that we should have clarified this in text. It is quite obvious that the tree structure strongly favors deletion of the phage element from 8B1 and not the other way around. This is clear in the original unrooted tree and in the rooted tree (Figure R1), both of which show that 8B1 is not the closest isolate to the LCA. This is also supported by the SNV divergence table (Figure R2) and is consistent with the fact that 8B1 was indeed the last *B. hinzii* isolate collected as a part of this study. We have modified the text appropriately in lines 129-130.

And again in lines 119-123 and Suppl Methods. They seem to have gone for an ad-hoc method of recreating an LCA, rather than using a robust phylogenetic approach with a rooted tree.

Response: We appreciate this feedback. While the text may not have been clear enough on this point, we actually did use multiple outgroup genomes in the construction of the original LCA. Furthermore, our choice of the LCA reconstruction approach was driven by underlying data structure. It is not a universal

approach, but we believe it is an effective approach for closely related isolates that diverged independently. Indeed, after accounting for the SNVs shared in the abundant DnaQ E9G clade, only two mutations are shared by more than 50% of isolates (Fig R1 and Fig R5). Below we explain the justification for the approach we used in greater detail, and also construct an LCA from the rooted tree as the reviewer suggests. The LCAs constructed by both methods turn out to be identical. We include both methods in the revised manuscript and have revised the text as indicated.

Figure R5. Frequency histogram of SNVs across full set of *B. hinzii* isolates. SNVs were identified relative to LCA and then numbers of isolates harboring each SNV were tabulated and plotted on the graph.

There are several reasons why we chose the custom approach. First of all, the procedure is simple and it allows non-specialists to appreciate underlying data analysis fully. Second, it allows straightforward reconstruction of a complete genome. Reconstruction of the LCA genome with maximum-likelihood approaches intrinsically involves construction of a core genome subset and genomic regions found only in the patient isolates will be missing. Third, commonly used maximum-likelihood tools are poorly suited for handling indels, such as found in the 1G1 isolate. It is certainly possible to design/adapt an approach utilizing a maximum likelihood framework but given the particular data structure of our dataset we believe that our procedure is simple, accurate and efficient.

To respond to the reviewer's point, we reconstructed the LCA with a rooted phylogenetic tree. To do this, we performed an independent reconstruction of the ancestral states for all genes of the core genome using PREQUEL from the PHAST v1.4 package using an ML phylogenetic tree built with all the

patient isolates and FDAARGOS_621 as an outgroup (Fig R1). Comparison of the most likely ancestral state reconstructed at the LCA node (indicated by red dot in Fig R1) with the one produced by our method using QUASt v5.0.2 showed no differences. We must note, however, that the comparison was restricted only to regions recovered in the LCA reconstructed from the rooted ML phylogenetic tree, as the ML-reconstructed LCA lacks the genomic regions that are common to all patient isolates, but absent from the outgroup. Since the ML LCA is identical to the original we constructed (with this qualification), we are highly confident that our LCA reconstruction procedure was robust and error-free. The ML LCA reconstruction and validation of our approach have been added to the revised manuscript in the Supplemental Methods section, lines 291-344.

The discussion of “two ancestors” from line 126 is also confusing. If this is assumed to be a single infection there will only be one ancestor of the whole infection, but the high-divergent strains will share a LCA within that tree.

Response: We agree with the reviewer that using a “two ancestors” description is unnecessarily confusing. Our assumption is that there is a single infection. By two “ancestors” we were referring to the LCA for the whole tree (ancestor 1) and the LCA for the subset of DnaQ E9G isolates (ancestor 2). We believe the DnaQ E9G clade ancestor (ancestor 2) descended from the “true” LCA, ancestor 1. To resolve this confusion, we have revised the text throughout so that only the LCA for the whole tree is referred to as the ancestor, and what was referred to as “ancestor 2” is now referred to only as the “DnaQ E9G clade founder”. We believe this rewording resolves this confusion and clarifies the text.

*Lines 131 to 139. The fact that isolates from mixed morphology colonies are more similar to each other than isolates taken on different days could be explained by there being geographically separate foci of infection that are evolving independently. This behavior has been reported using deep sequencing in chronic lung infections such as *Pseudomonas aeruginosa*. This should be discussed.*

Response: We agree with the reviewer that we cannot rule out spatially separated populations evolving independently as contributing to the similarity of the isolates from mixed morphotype colonies, though we do not have any direct evidence for this. We have edited the text to include this possibility explicitly, along with an appropriate reference to the cystic fibrosis literature where this has been examined in more depth (lines 600-603). We thank the reviewer for this suggestion.

Line 175. I would disagree that hypermutation itself has a selectable advantage. It is certainly the case that variants with a selective advantage arise more rapidly on a hypermutator background, but the hypermutation variant does not itself confer a selective advantage. Selection does not have foresight. A variant cannot be selected because it will have an advantage at a later date. The hypermutator variant increases in the population because it hitchhikes on the selective variants that arise later. This also means that these variants are not “co-selected” with the hypermutation variant (Line 177). They are the primary selected variants, and the hypermutation variant is a hitchhiker.

Response: We agree with the reviewer's points, and we appreciate that the writing was unclear. We have rewritten the relevant sections in response, including lines 235-241 and 469-482.

This is expanded in the discussion (line 308), with the concept that additional hypermutator lesions enhance "adaptive evolutionary searches". Again, selection has no foresight. A mutation cannot be selected because it will provide a benefit in the future (after the search). It may appear to be the case after the fact, but that is only because it has hitchhiked to higher frequency on the back of selective mutations that have occurred on that genetic background. This may appear to be a semantic argument, but I think it is important to be clear about selective processes.

Response: We agree and have removed this language and revised all of the relevant sections accordingly as above.

Line 177. Positive selection is not indicated by the number of non-synonymous changes. Mutations arise randomly, and non-synonymous variants arise more frequently than synonymous ones because they have more opportunity to do so (there are more sites in the genome where non-synonymous mutations can occur). To show positive selection, the authors must calculate: (number of non-synonymous variants per non-synonymous site)/(number of synonymous variants per synonymous site); i.e. dN/dS; and then show that it is significantly different from the neutral expectation.

Response: Yes, we certainly agree with the reviewer's point that the number of non-synonymous mutations is indicative of positive selection only if it is significantly different from the neutral expectation. We have now performed explicit genome-wide dN/dS calculations as suggested by the reviewer (explained in Methods, lines 759-766). Overall, we find evidence of mild purifying selection genome-wide in the patient isolates, with a dN/dS = 0.81. The genome-wide dN/dS ratio is higher in isolates evolved in the mutational accumulation experiments (dN/dS = 0.91, P=0.042, Fisher's Exact Test) suggesting weaker selective pressure under conditions of *in vitro* growth in rich media. Within the patient isolate set, multiple genes (many of which also have a significant excess of independent mutations) do have elevated dN/dS ratios indicating that they are under positive selection. We have included this additional information in the manuscript in lines 243-248, 353-357, and 478-479 and Table S8.

When we perform dN/dS analysis for the 24 CDS substitutions in the E9G clade founder, 22 of them are non-synonymous leading to a dN/dS ratio of 3.51, providing evidence of positive selection. This dN/dS ratio is significantly higher than the overall genome-wide dN/dS ratio in patient isolates (P< 0.016, Fisher's Exact Test). Thus, although our original description was largely empirical, we now present better evidence of positive selection in the DnaQ E9G clade founder and have included this evidence with discussion in the revised manuscript in the sections indicated above. We thank the reviewer for this suggestion, which improved the manuscript.

The section beginning line 220 is confusing. The authors say they are looking for genes which were “independently mutated in different patent isolates”. However, the simulation described in the supplementary material is a permutation test looking at mutation burden in each gene. It does not test whether those mutations occurred independently in different isolates (i.e. convergent evolution). For that, they would need to do a homoplasy test using a phylogenetic tree, which they do not describe. They should change the description or do a homoplasy test (or both).

Response: We agree with the reviewer that the description was somewhat unclear. To clarify the approach for the reviewer: The simulation includes only the non-redundant set of unique mutations; that is, we include in the analysis independent mutational events. Some of these mutations were present in multiple isolates, but the majority, 3913 out of 6040 mutations were found only in a single isolate (Figure R5). SNVs seen in multiple isolates were in most cases unambiguously explained by shared ancestry. Regardless of the number of isolates where each of the SNVs was detected, it was counted as a single event in our simulation.

To further evaluate consistency of our SNV dataset with the phylogenetic tree a homoplasy test was performed (Methods below). In total, 85 out of 6459 mutations were not explained by shared ancestry. These mutations may indeed have arisen independently multiple times, but alternative explanations include recombination and artifacts associated with tree assembly from very sparse data. In any case, the percentage of identified homoplastic mutations out of the total set of mutations was very low (1.3%), and 96.1% of the shared mutations are explained by shared ancestry, consistent with their independent origin. We believe this justifies our approach of using the non-redundant set of unique mutations for the purpose of the simulation, and in fact believe this is the more conservative approach. We would expect the effect of the addition of the homoplastic variants would be to increase the significance of the findings, as it would increase the mutation counts for the genes with which they are associated.

Although we estimate that very few mutations emerged repeatedly, many of the genes (e.g., *bvgS*) had as many as 10-12 multiple different mutations – different positions, different substitutions - in different isolates, significantly more than expected by chance. The goal of the analysis was to detect those genes with excesses of mutations or higher mutational burden. The fact that given gene is repeatedly and independently mutated in the set of patient isolates indicates that it is likely under positive selection. Given the fact that the mutation set is very sparse and 96.1% of shared mutations are explained by shared ancestry, we believe that our simulation provides a fair estimate of the expected number of mutations in each gene and allows us to evaluate statistical significance for the excess of mutations per gene. We agree with the reviewer that the description should be improved and we have clarified the description in appropriate places (lines 348-365) in the revised manuscript.

Minor points:

Line 227. Please correct “mutated x times”

Response: We have corrected this error in the revised manuscript (“mutated 10 times”).

Methods used for responses to Reviewer #1:

Phylogenetic analysis and LCA reconstruction: In order to compare our method of LCA genome reconstruction to a maximum likelihood approach, we first assembled a core genome. To do so, we generated SNV-substituted complete genome sequences for each isolate using the consensus outputs from Snippy v4.4.1 (<https://github.com/tseemann/snippy>). Raw sequencing reads from each patient isolate were used to call SNVs using the complete PacBio genome of 2B3 isolate as a reference. These SNVs identified by Snippy were then applied to the 2B3 reference genome sequence to produce a reconstructed complete genome sequence for a given isolate. Those sequences were then used together with the complete genome of FDAARGOS_621 to reconstruct an aligned and concatenated core genome. A similarly organized core genome was also produced using the LCA genome that was constructed with the method from our manuscript for comparison purposes. The core genome of the patient isolates and of the outgroup were then used to reconstruct a phylogeny using RaxML v8.2.12 (ref). The aligned sequence and the reconstructed tree were then used Fto reconstruct the most likely ancestral state, by using phyloFit from the PHAST v1.4 package to first fit the tree model to the multiple alignment of core genomes sequences by maximum likelihood, using the GTR substitution model. Then the tree model was used in conjunction to the sequences with PREQUEL from the PHAST package to reconstruct the most likely ancestral state at the node corresponding to the LCA (fig R1). PREQUEL computes marginal probability distributions for bases at ancestral nodes in the phylogenetic tree. However because of the difficulty of dealing with indel events, they are not treated probabilistically but rather reconstructed by parsimony. Finally, the reconstructed sequence was compared to the ancestral core genome reconstructed using the method described in our manuscript using QUASt v5.0.2. All events identified by QUASt within 5 bp from the end or the beginning of genes were ignored as they corresponded to expected artefacts created by the artificial gene junctions in the concatenated core genome.

dN/dS calculation: To compute dN/dS, the number of synonymous sites and non-synonymous sites was calculated for each gene using gene annotation information from the gff files. In order to evaluate the difference in proportions of non-synonymous substitutions, we aggregated data either for the whole genome, or for the 24 genes containing mutations in DnaQ E9G clade founder.

Homoplasy test: To look for homoplasy, the R package homoplasyfinder v 0.0.0.9000 (<https://doi.org/10.1099/mgen.0.000245>) was used. A tree was reconstructed in RaxML v8.2.12 by using the concatenated core genome of all the isolates. The tree was then compared to a table containing the information of presence and absence of the variants for the different patient isolates. Variants with a consistency index below 1 were identified as variants conflicting with the topology of the tree, which could be explained by more than one mutational event of recombination.

Reviewer #2 (Remarks to the Author):

Launay A et al. described in vitro characterisation of a emergent pathogen, Bordetella hinizzi, found in a patient with an inborn error of immunity (IEI). Following a period of 45 months and using a clonal lineage, the authors detected a diversification of genetic in the bacteria. The description of microbiological procedures is very well documented. However the link between the microbe and the host is not clear. Several data concerning IL-12Rβ1 deficient patient are missing.

Response: We thank the reviewer for the multiple suggestions. In response, we have added a number of additional details regarding the clinical history of the patient, as detailed below, and we believe this has improved the manuscript. Note that the line numbers given below refer to the revised manuscript.

B. hinizzi can be a cause of infectious disease in human (endocarditis, bacterimia, pneumonia ...). Why the authors followed the diversity of phenotype in two samples : blood and stool ? they had access to other samples ? B. hinizzi infection was localized or disseminated ?

Response: *B. hinzii* was isolated only from blood and gastrointestinal cultures in this patient. It was not isolated from respiratory cultures that were performed. We cannot rule out that the patient had infection at other sites outside of the intravascular space and the gastrointestinal compartment (eg lymph nodes, etc), but these are the two culture types the clinical laboratory received and from which the organism was isolated. We would consider the infection to be “disseminated” based on the fact that the patient had persistent (45 months) infection in both the intravascular and gastrointestinal compartments. We have clarified this point in the text.

What is the infectious history in the IL-12Rβ1 patient ? it includes other pathogens found in IL-12Rβ1 deficient patients, like salmonellosis ?

Response: The patient had a history of *C. tropicalis* esophagitis. We are not aware of any history of either mycobacterial infection or salmonellosis. The patient has had a long history of clinical colitis, and it is not clear whether this is related to, or caused by, the *B. hinzii* GI infection. The patient also has had a history of membranoproliferative glomerulonephropathy and consequent nephrotic syndrome with proteinuria. We have added this information to the text in lines 97-100.

what is the mutation ?

Response: The patient is homozygous for IL12RB1 c.94C>T p.Gln32Ter. We have added this information to the text in line 95. The patient is also heterozygous for a ABCB1 c.3266C>T p.Pro1089Leu mutation. We do not know whether this latter mutation contributes to his disease phenotype, though in other

patients, mutations in ABCB1 may be associated with colitis. However, because of the uncertain role of this latter mutation in the disease phenotype (if any), we have not included it in the manuscript.

In general this IEI has an incomplete penetrance of mycobacteria and salmonellosis. I am curious to know if other members of the family carried the mutation in the same state of patient and if the individuals ave also a diversity of genetic in B. hinizzi.

Response: The parents are both heterozygous for IL12RB1 c.94C>T p.Gln32Ter and we are not aware of a history of infections in them.

In the discussion, the authors claim that some IEI have selective predisposition to some microbes. I agree with this comment. However, the selection of IEI is not clear, in particular idiopathic CD4 lymphopenia which by definition none genetic mutations were found. What is the connection around the cited IEI ?

Response: We agree that CD4 lymphopenia is not an IEI, but it was included as an example to illustrate that many immunodeficiencies (not necessarily strictly IEIs) predispose to infection with certain genera of bacteria, which are able to exploit specific deficiencies in host immunity. We have reworded line 501 to clarify that the list included both inherited (genetic) and acquired immunodeficiencies.

Minor comments

Patients with IL-12Rb1 deficiency have a reduced levels of IL-12/IL-23 stimulated IFN- γ production. The stimulation with both cytokines is affected (second paragraph of introduction).

Response: We thank the reviewer for pointing this out and have rewritten the text in line 81 to reflect this.

REFERENCES

Gurevich A, Saveliev V, Vyahhi N and Tesler G, QUASt: quality assessment tool for genome assemblies, Bioinformatics. 2013; 29 (8):1072-1075.

Page AJ, Cummins CA, Hunt M, Wong VK, Reuter S, Holden MT, et al. Roary: rapid large-scale prokaryote pan genome analysis. Bioinformatics. 2015;31(22):3691-3.

Hubisz MJ, Pollard KS, Siepel A. PHAST and RPHAST: phylogenetic analysis with space/time models. Brief Bioinform. 2010; 12(1):41–51.

Stamatakis A. RAxML version 8: a tool for phylogenetic analysis and post-analysis of large phylogenies. Bioinformatics. 2014;30(9):1312-3.

Reviewers' Comments:

Reviewer #1:

Remarks to the Author:

I am happy that the authors have fully addressed my comments.

Reviewer #2:

Remarks to the Author:

Launay A and colleagues have incorporated all the suggestions made by both Reviewers into the manuscript. The manuscript will be helpful for understanding the evolution of pathogens in immunodeficient patients with chronic infectious diseases.